# Effect of Microstructural Evolution on the Mechanical Properties of Ni-Cr-Mo Ultra-Heavy Steel Plate

**DOI:** 10.3390/ma16041607

**Published:** 2023-02-15

**Authors:** Kaihao Guo, Tao Pan, Ning Zhang, Li Meng, Xiaobing Luo, Feng Chai

**Affiliations:** 1Division of Structural Steels, Central Iron and Steel Research Institute, Beijing 100081, China; 2Metallurgical Technology Institute, Central Iron and Steel Research Institute, Beijing 100081, China

**Keywords:** microstructural evolution, ultra-heavy steel plate, multivariate function, EGS, M-A constituent

## Abstract

In this study, microstructural evolution and its effects on mechanical properties across the thickness of a 120 mm Ni-Cr-Mo industrial ultra-heavy steel plate were quantitatively investigated by means of optical microscope (OM), scanning electron microscope (SEM), transmission electron microscope (TEM) and electron back-scatter diffraction (EBSD). The results show that the martensite fraction is 65% at 10 mm and disappears at 40 mm, while granular bainite appears at 35 mm and climbs up to as high as 32% at 60 mm, with M-A constituents significantly coarsened. The strength drops with the gradual coarsening of the laths as well as decreased martensite fraction from the surface to the centre. The toughness is mainly affected by the block size and the morphology and quantity of M-A constituents. This study established a multivariate function between the microstructure and toughness (50% fibre area transition temperature, FATT_50_) with careful consideration of the influence of effective grain size (EGS) and M-A constituent size distribution.

## 1. Introduction

Rapidly developing large-scale offshore platforms highly demand a high-strength, high-toughness ultra-heavy steel plate with a stable and consistent microstructure through a full-thickness section [1,2]. Ni-Cr-Mo-V multi-alloying hardenability enhancing and high-pressure roller quenching followed by tempering (Q-T) are used to produce heavy plates with a fully tempered martensitic/bainitic microstructure [3,4,5]. However, it is rather challenging to obtain full martensite across the thickness section for increasing plate thickness, especially for that beyond 100 mm.

The different cooling rates across the thickness under the quenching process is the main factor that affects the microstructural homogeneity [5,6,7]. In general, a heavy plate under the roller quenching process mainly consisted of three types of microstructure: martensite (M), martensite/lath bainite (M/LB) and lath bainite/granular bainite (LB/GB) [8,9,10]. Tempered martensite usually has an excellent combination of strength and toughness [11]. The properties of lath-like M or M/LB are determined to a large extent by their substructure size, which forms at different cooling rates [12]. Quarter thickness position tends to form an M/LB mixed microstructure, rather than a unitary M microstructure, due to a slower cooling rate than the surface, and always accordingly presents a lower ductile-brittle translation temperature than the surface [13,14]. However, the centre of the heavy plate sometimes exhibits a GB component if the cooling rate of the position is insufficiently fast. The microstructure with a GB component frequently and seriously deteriorates low-temperature toughness since the nucleation and propagation of the microcrack are often observed around martensite/austenite constituents (M-A) within GB [15,16,17]. It is generally accepted that the size of M-A constituents increases with the decreased cooling rate and results in decreased toughness. Recent studies clarified that the toughness of full thickness often presents “M”-shape characteristics, and the toughness at the surface and 1/2 position were reduced by the coarse martensite carbides or granular bainite [18]. However, most of the current studies for heavy plates are limited to the position of the surface, 1/4 and 1/2, which is helpful for the understanding of the characteristic of the heavy plate but is insufficient for the analysis of the microstructural evolution. It is still significant to probe the evolution of the microstructure and properties across the thickness of the heavy plate.

In the present work, a quantitative analysis of microstructure evolution and its correlation with properties across the full thickness of a 120 mm Ni-Cr-Mo industrial heavy plate was studied. In particular, the toughness and strength evolution through the thickness was discussed in detail, and the multivariate function between microstructures and FATT_50_ was established.

## 2. Materials and Methods

The investigated material was carried out by Ni-Cr-Mo steel with a thickness of 120 mm industrially produced in order by casting, electroslag remelting, rolling and Q-T heat-treatment. The as-rolled plate was reaustenitised at 900 °C and was subsequently quenched using an industrial roller to room temperature and then tempered at 600 °C. The quenching and tempering soaking times are determined by the thickness of the steel plate, 2.5 min/mm and 5 min/mm, respectively. The chemical composition of the investigated steel is shown in Table 1. In order to assess the properties and microstructures across the section of the plate, the specimens were cut at different locations, as shown in Table 2. The property and microstructure of the heavy plate are thought to be symmetrical, and the positions selected from the surface to the centre of the heavy plate represent the full thickness.

The specimens etched in 4% nital solution were observed by a scanning electron microscope (SEM, model: Quanta650 FEG, FEI Company, Hillsboro, OR, USA). The prior austenite grain boundaries were etched by a saturated picric acid reagent and observed with an optical microscope (OM, model: Leica MEF4M, Leica Company, Wetzlar, Germany). Electron back-scatter diffraction (EBSD) analysis (step size: 0.2 μm) was performed by a field emission scanning electron microscope (FE-SEM, model: JSM-7900F, JEOL Company, Tokyo, Japan), and the data were analyzed with HKL CHANNEL 5 Flamenco software. The ultrafine laths were investigated by a field emission transmission electron microscope (H-800, 200 KV) using the thin foils electro-polished at −20 °C in a 6% perchloric acid solution. The density of dislocation was measured by an X-ray diffractometer (XRD, model: D8 ADVANCE, Bruker Company, Bremen, Germany) operated at 35 kV, and the specimens were handled at a scan rate of 2°/min.

The tensile strength of the experimental steels was measured at 20 °C with a gauge length of 25 mm and diameter of 5 mm. The tensile direction is perpendicular to the rolling direction. The impact tests were performed with standard Charpy V-notch specimens at temperatures ranging from −20 °C to −120 °C. Three specimens were used per condition to avoid deviation. Instrumented impact test was carried out at −60 °C. Vickers hardness measurements were carried out under an applied load of 10 kg.

## 3. Results

### 3.1. Microstructure

Figure 1 presents the SEM micrographs across the section of the heavy plate, showing a microstructural transition of M to LB and GB from the surface to the centre. It can be observed that the steel at 10 mm mainly consists of a tempered martensitic microstructure. From 20 mm to 30 mm, the microstructure is mainly composed of LB and a small fraction of martensite. Whereas, from 40 mm to the centre, the microstructure consists of LB and GB. Specimens at 10, 30 and 60 mm were selected to reveal the key information about the three typical microstructures (M, M/LB and LB/GB) in the following study. Figure 2 shows the morphology of a prior austenite grain (PAG) at three typical positions. According to the OM images, some PAGs severely coarsened as the distance from the surface increased, especially at 60 mm. The average PAG size of each specimen was calculated based on the measurements of more than 200 PAGs, and the detailed results are shown in Table 3.

Figure 3 presents the TEM images of the three specimens. It is observed that the specimen at the sub-surface (10 mm) of the heavy plate shows a typical fine martensitic lath with cluttered carbides (Figure 3a). At 30 mm, the microstructure consists of coarse laths and rod-like carbides with an intersection angle of about 60° (Figure 3b), which is a typical LB structure [19]. Similar to the specimen at 30 mm, the microstructure of the specimen at 60 mm involves coarse bainitic laths and carbides. The width of the bainitic lath is obviously coarser. Some M-A constituents within bainitic laths are also found for the specimen at 60 mm (Figure 3c,d). The carbides of the specimen at 10 mm are smaller than that at other positions. There is no significant change observed from TEM maps in the carbide size after 10 mm.

The EBSD grain boundary maps are shown in Figure 4a–c. The EBSD maps reveal that the fraction of the low-angle grain boundary (LAGB) is greater on the surface than at other positions of the plate, but the fraction of HAGB (the boundary angle larger than 15°) is greater in the quarter of the plate. Figure 4d illustrates the misorientation angle distribution of grain boundaries, including the LAGB and the HAGB. The sum of grain boundary density decreases across the section, as shown in Figure 4d. The majority of changes of HAGB are those larger than 55°; these boundaries firstly increase from 0.502 μm^−1^ (at 10 mm) to 0.849 μm^−1^ (at 20 mm) and then decrease to 0.382 μm^−1^ (at 60 mm). The effective grain size (EGS) obtained from the EBSD results is presented in Table 3. The EGS is the average size of grains separated by HAGBs. The EGS of the specimen at 20–30 mm is relatively smaller than that at other positions.

The PAG will be divided into a set of different structural units, such as packets, blocks and laths, which are the key structural parameters controlling the properties. At least 100 packets, 500 blocks and 200 laths were measured in order to ascertain the average packet size, block width and lath width. The packets, blocks and laths were characterized by SEM, EBSD and TEM, respectively [13]. The average size of the substructures is given in Table 3. The results present that the packet size increases from 9.72 μm (at 10 mm) to 12.94 μm (at 35 mm) while the block width first decreases and then increases through the thickness of the steel. The substructure size was not determined after 35 mm, where a large fraction of GB was observed.

To further illustrate the evolution of martensite fraction through the steel thickness, band contrast (BC) numbers selected from the EBSD data were employed. The current research proves that martensite has a lower Kikuchi pattern intensity resulting from greater lattice imperfections compared with bainite, which leads to a lower BC value [9,20]. In this study, the threshold value takes advantage of the BC curve’s differential extremum, considering that the value may vary at different test conditions. The average statistical proportions of the martensite phase are shown in Figure 5. After 40 mm, the BC curve and differential curve all show a single peak shape, indicating the disappearance of martensite.

The GB fraction was calculated through the SEM maps, and it increased from 6% (35 mm) to 32% (60 mm), as shown in Table 3. Quantification of the M-A constituent is presented in Figure 6. The M-A constituent size continually increases from quarter to centre across the thickness direction. The M-A constituents at 50–60 mm are significantly concentrated on a large scale rather than those at 35–40 mm, according to the distribution curve of the maximal size (L_max_) of the M-A constituent fitted by the Weibull function, as shown in Figure 6a. The statistic histogram also presents an increasing amount of large-size M-A constituents at 50–60 mm, as shown in Figure 6b. The average maximal sizes are presented in Table 3.

### 3.2. Mechanical Properties

The mechanical properties of the specimens shown in Figure 7a present that the strength of the steel gradually decreases across the section, especially for yield strength from the position from 10 mm to 35 mm. The fibrous area fraction of the fracture at different test temperatures was measured and fitted by the Boltzmann function, and the FATT_50_ was selected as the ductile-brittle transition temperature. It is observed that the FATT_50_ presents a “V” shape across the thickness, as shown in Figure 7b. The FATT_50_ of the test steel rapidly decreases from the sub-surface to the 20 mm position and then continually increases with the increase of thickness.

## 4. Discussion

### 4.1. Microstructural Evolution through Thick Section

The reduced rolling strain along the thickness of the steel results in decreased nucleation sites and poor strain energy, which subsequently influences the PAG size. Meanwhile, the decrease in cooling rate from the surface to the centre eventually led to an evolution of microstructure. Figure 8 presents a numerical evolution of microstructure fraction through the thickness. The martensite disappears at 40 mm, and GB appears at 35 mm. The steel hardness is closely related to the martensite fraction, and it first decreases from the surface to 40 mm and then remains a constant.

The sub-structure size and HAGB play a very important role in the properties. HAGB density first increases and then decreases across the plate thickness, opposite to the change in block width. Long pointed out that the misorientation angle of 45–55° and 55–65° originate from the packet and block boundaries, respectively [21]. The increase in HAGB is highly related to finer block size. HAGBs are significantly influenced by the variant’s selection in austenite. The single austenite grain may produce 24 variants during transformation according to the Kurdjumov–Sachs (K–S) orientation relationship. These 24 variants can be divided into three Bain groups and four CP groups. The packet boundary is the boundary between variants in different crystal packet groups, and the block boundary is the boundary between variants in different Bain groups.

The inter-variant boundary of the V1 and V4 variants pair and the V1 and V2 variants pair is about 10° and 60°, respectively. Figure 9 contains the variants selection feature in austenite at different thickness positions. The V1 and V4 variants pair take the main group at 10 mm and 60 mm. Meanwhile, there are large amounts of V1 and V2 variants found at 30 mm, which results in an increased density of HAGBs and a decreased block width. It is considered that the bainite that transformed at lower temperatures needs more V1 and V2 variants to adjust the transformation strain, while the bainite transformed at higher temperatures adjusts by the plastic deformation of austenite [22,23]. It is widely accepted that the first-generated lath bainite isolates the prior austenite grains and subsequently hinders the growth of martensite [24,25]. Therefore, the microstructure at a position of 20–30 mm, which is composed of martensite and low-temperature-transformed lath bainite, has a large fraction of HAGBs, especially those exceeding 55°.

### 4.2. Correlation between Strength and Microstructure

It is well known that the influence of microstructure on strength mainly included fine grain strengthening, precipitation strengthening, dislocation strengthening, etc. The dislocation density was calculated to be 3.95 × 10^14^ m^−2^, 3.26 × 10^14^ m^−2^ and 3.04 × 10^14^ m^−2^ at the position of 10 mm, 30 mm, and 60 mm, respectively, using XRD. The dislocation strengthening is about 151 MPa, 137 MPa and 133 MPa, respectively. The precipitation strengthening can be expressed by the Orowan relationship [26,27]. In this study, the average carbide sizes are 54.5 nm, 52.1 nm and 74.3 nm at 20 mm, 40 mm and 60 mm, respectively. After the same high-temperature tempering, the average size and distribution of carbides are barely modified beyond 20 mm, and the value of precipitation strengthening can be estimated for a range from 154 MPa to 143 MPa. The precipitation strengthening and dislocation strengthening all slightly decrease across the plate thickness. The following Hall–Petch relation was used to estimate the effect of the grain size on the strength [26,27]:σ_g_ = k_y_d^−1/2^,(1)
where σ_g_ is the grain boundary strengthening, ky is the Hall–Petch slope, and d is the effective grain size.

The microstructure mainly consists of a lath structure across the section of steel, especially for the position from the surface to 35 mm. The substructure size is the controlling unit for the properties, and their respective contribution to the strength has not achieved a general consensus. It takes a long time to take the block as the controlling unit for strength [28]. However, previous studies also point out that lath boundaries can provide barriers to dislocation motion [29,30]. Figure 10a presents the correlation between several sub-unit sizes and material strength. The yield strength within 10–35 mm decreases from 915 MPa to 851 MPa, while the laths width increases from 0.30 μm to 0.52 μm. There is a significant linear relationship between d_lath_^−1/2^ and strength, as shown in Figure 10b. In fact, the coarsening of the lath across the thickness results from a decrease in the martensite fraction, as shown in Figure 10c,d, which depicts a linear relationship between strength and martensite fraction across the thickness. It is accepted that the decreased martensite fraction results in coarse lath and decreased strength. Thus, an increase in the GB fraction within LB/GB microstructure at the position after 40 mm may also result in a slight decrease in strength.

### 4.3. Correlation between Toughness and Microstructure

The influence of the sub-structure size on the toughness presented in Figure 11a. The evolution of block size is regarded as the primary factor influencing low-temperature toughness; however, it does not mean that the packet boundary has no contribution to the toughness. Figure 11 shows the correlation between the substructure orientation and crack propagation, with the white line representing angles between 15 and 55 degrees and the black line representing angles greater than 55 degrees. The crack was found to be deflected at both packet boundaries and block boundaries. However, the deflected angle of crack propagation at packet boundaries is larger than that at block boundaries.

However, considering the multi-type microstructure observed, it is relatively difficult to establish a quantitative relationship between substructure size and low-temperature toughness across the thickness of the plate. Therefore, effective grain size, which may demonstrate crystallographic features for both M, LB, and GB, was employed to examine the quantitative correlation further [31,32]. The following equation can be used to describe the quantitative relationship between EGS and FATT_50_:FATT_50_ = T_0_−KD^−1/2^,(2)
where K is the structural constant representing the influence of grain size on toughness, and D is the effective grain size.

It is well known that the growing grain size results in the rise of FATT_50_. However, the increase in the granular bainite also contributes to the rise of FATT_50_ [15]. The damage of GB to toughness is not only owed to the coarse grains but also owed to the presence of large-scale M-A constituents [32,33]. Figure 12 presents that the secondary cleavage cracks are normally initiated at M-A constituents, confirming the role of M-A constituents in the brittle cleavage fracture of LB/GB.

However, there are rare models provided to quantify the effect of the M-A constituents on the toughness. Dewei Tian proposed a model which takes into account the grain size, M-A width and interspacing [34]. S. Lee studied the quantitative influence of the carbides interspacing on the toughness and then established a relationship between K_Jc_ and −1/4 power of carbides quantity, which was also deemed as the brittle structure [35]. Considering the studies above, the number of M-A constituents larger than the critical crack size (D_c_) is selected as a parameter in the model. The critical crack size is obtained by the Griffith function:(3)$$\sigma_{c}={\left(\frac{{\pi E\gamma }_{p}}{\left(1-\nu^{2}\right)d}\right)}^{1/2},$$
where γ_p_ is the effective surface energy of the microcrack, ν is the Poisson’s ratio, E is Young’s modulus, d is the length of the critical crack and $\sigma_{c}$ is the critical cleavage stress. Then, relevant literature shows that the critical cleavage stress relates to the general yield load (Py) and can be expressed from the following equations:(4)$$\sigma_{yd}=467P_{y}/B.$$

The critical size can be determined to be 1.6 μm based on the results of instrument impact performed at −60 °C.

The number of M-A constituents larger than the critical crack size is estimated using the following equation, which takes into account the GB fraction and M-A density:(5)$$N_{A}=GB\%\cdot Density_{M\mathrm{-A}}\cdot P\left\{L_{max}>D_{c}\right\},$$
where *P*{*L_max_ > D_c_*} was calculated using the Weibull distribution of M-A constituents in Figure 6a.

Then, a comprehensive fitting model to characterize the correlation between the microstructure and FATT_50_ is presented as follows:FATT_50_ = −108*EGS^−^*^1/2^ + 903 × (*GB%* × *M-A Density* × *P*{*Lmax > D_c_*}) − 15 °C.(6)

The R^2^ of the fitting equation is approximately 0.99, which does not take the position of 10 mm into account. It has widely observed a deteriorated toughness at the surface, which partially exists on the surface of the heavy industrial plate and involves numerous parameters, such as carbides or residual stress [4,36]. The relevant analysis will be presented later and will not be explained in this article. The relevant calculated values are shown in Table 4.

The correlation between the microstructure and FATT_50_ is shown in Figure 13. It has been widely recognized that the application of the Hall–Petch relationship in M/LB. For the type of GB/LB, a large fraction of GB appeared to take the grain size increase significantly, and M-A constituents inside increase the tendency of brittle cracking. It is obvious that grain size plays a dominant role in low-temperature toughness. With the increase of the large-scale M-A constituents, the damage to the toughness also increases. Due to a limited percentage of the coarse M-A constituent, the harm that M-A constituents produced at 35 mm is relatively poor. However, at 60 mm, a significant number of coarse M-A constituents raise the FATT_50_ by 15 °C. Although the created fitting model is built on simple assumptions, it evaluates the harm caused by M-A constituents to material toughness and presents a more thorough correlation between microstructure and toughness.

## 5. Conclusions

In the present work, the evolution of microstructure and mechanical properties through the thickness of a 120 mm Ni-Cr-Mo industrial heavy plate were investigated, and the correlations between them were discussed in detail. Based on the results obtained in the current work, the following conclusions can be drawn:(1)The microstructure is primarily composed of three types, M, M/LB and LB/GB. The martensite fraction is 65% at 10 mm, and it completely disappears at 40 mm. The GB appears at about 35 mm and decreases to 32% in the centre. The average Lmax of M-A constituents increases from 0.49 μm to 0.7 μm. The HAGB density takes a peak value of 1.2 μm^−1^ at 20 mm, while the change of dislocation density is limited across the thickness of the plate;(2)The strength across the section nearly monotonously decreases with the coarsening of lath, especially for the position from 10 mm to 40 mm, which results from the decrease in martensite fraction;(3)The FATT_50_ across the section was mainly affected by the grain size and granular bainite fraction, which first increases and then decreases. This study established a multivariate function between the microstructure and FATT_50_, and the effect of grain size, GB fraction, and the distribution of M-A constituent’s size on toughness are taken into account in the relationship. The function is that FATT_50_ = −108d^−1/2^ + 903 × (GB% × M-A Density × P{Lmax > Dc}) − 15 °C.

## Figures and Tables

**Figure 1 materials-16-01607-f001:**
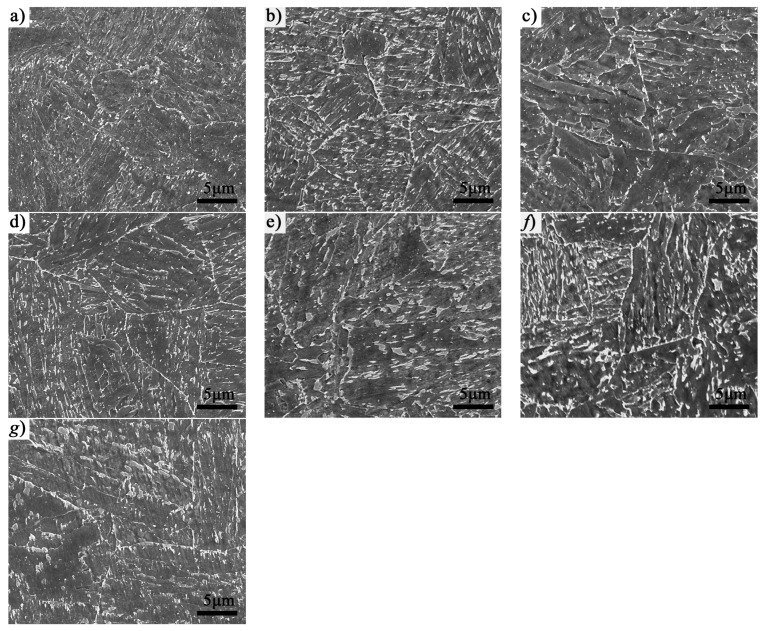
SEM graphs of the specimen at different positions: (**a**) 10 mm, (**b**) 20 mm, (**c**) 30 mm, (**d**) 35 mm, (**e**) 40 mm, (**f**) 50 mm and (**g**) 60 mm, which present a microstructural transition from M to LB and GB.

**Figure 2 materials-16-01607-f002:**
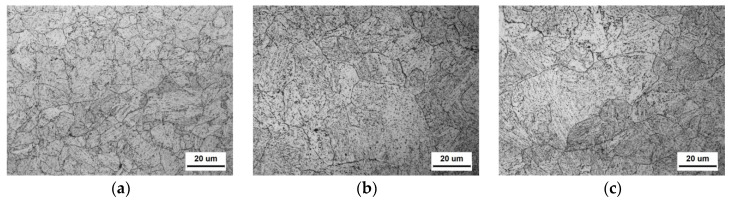
Prior austenite grain of the specimen at different positions: (**a**) 10 mm, (**b**) 30 mm and (**c**) 60 mm.

**Figure 3 materials-16-01607-f003:**
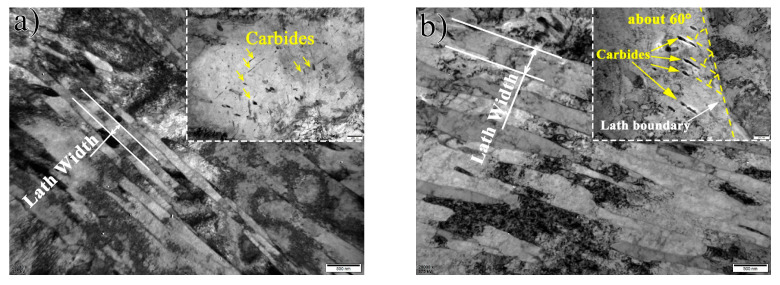
TEM presents the lath width and typical carbides distribution at different positions: (**a**) 10 mm, (**b**) 30 mm and (**c**) 60 mm. (**d**) Shows the M-A island observed at 60 mm, and the inserted figures in (**a**–**c**) present the evolution of carbides.

**Figure 4 materials-16-01607-f004:**
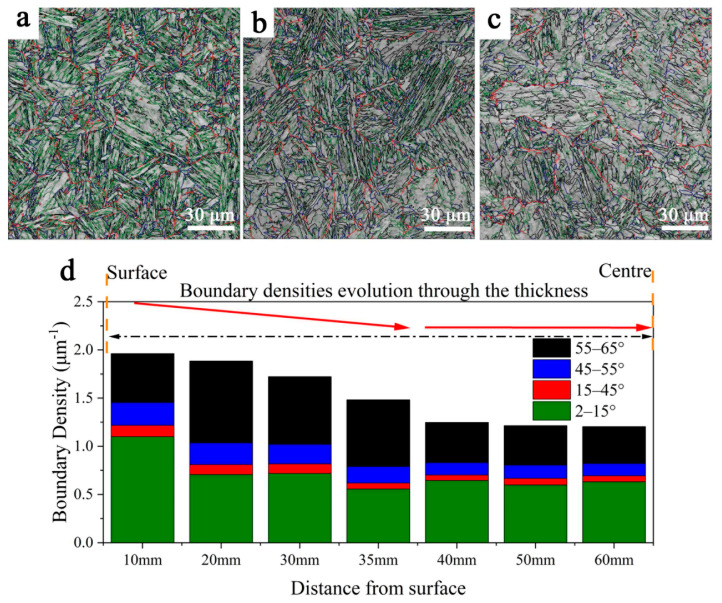
EBSD maps for different positions: (**a**–**c**) are grain boundary maps at 10 mm, 30 mm and 60 mm, and the boundary angles are 2–15° in green lines, 15–45° in red lines, 45–55° in blue lines and 55–65° in black lines, and (**d**) presents the evolution of the boundary density.

**Figure 5 materials-16-01607-f005:**
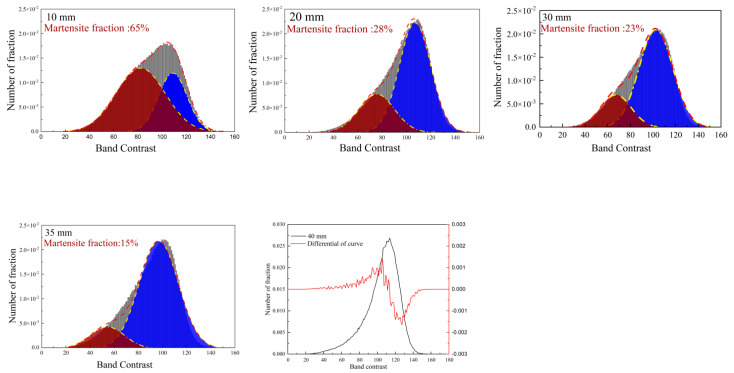
Band contrast value frequency and Gaussian fitting of specimens within 40 mm. The threshold corresponds to the BC curve’s differential extremum. The red colour peak corresponds to martensite and the blue colour to bainite.

**Figure 6 materials-16-01607-f006:**
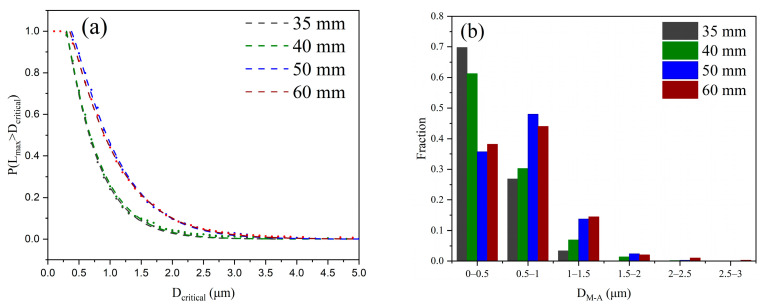
The maximum size of M-A constituents changed with different positions. (**a**) Line fitting using the Weibull function and (**b**) a statistical histogram.

**Figure 7 materials-16-01607-f007:**
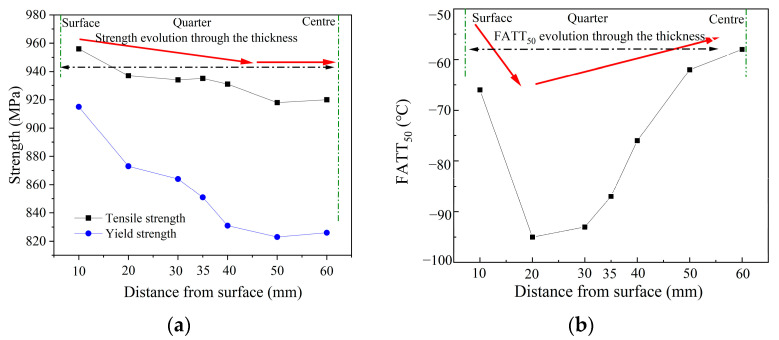
Mechanical properties of heavy plate: (**a**) strength evolution through the thickness and (**b**) evolution of FATT_50_ shows a “V” shape.

**Figure 8 materials-16-01607-f008:**
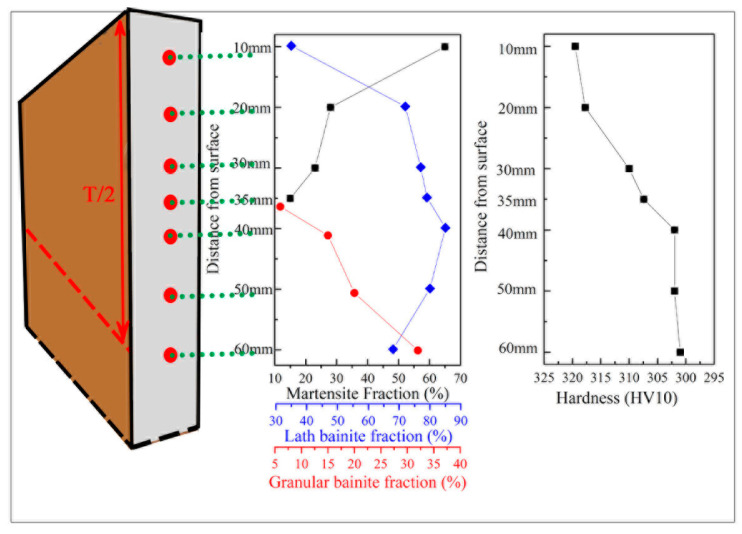
Microstructural evolution through the thickness of the heavy plate.

**Figure 9 materials-16-01607-f009:**
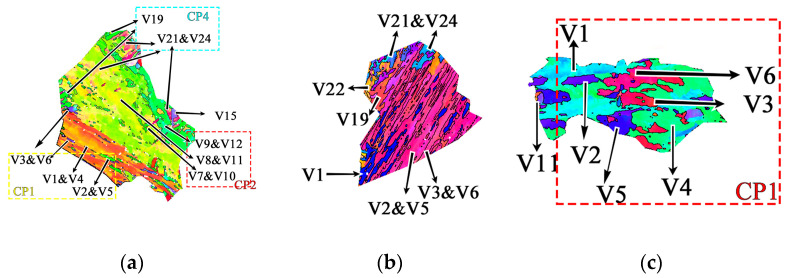
IPF maps showing variants feature within prior austenite grains reconstructed at the position of (**a**) 10 mm, (**b**) 30 mm and (**c**) 60 mm.

**Figure 10 materials-16-01607-f010:**
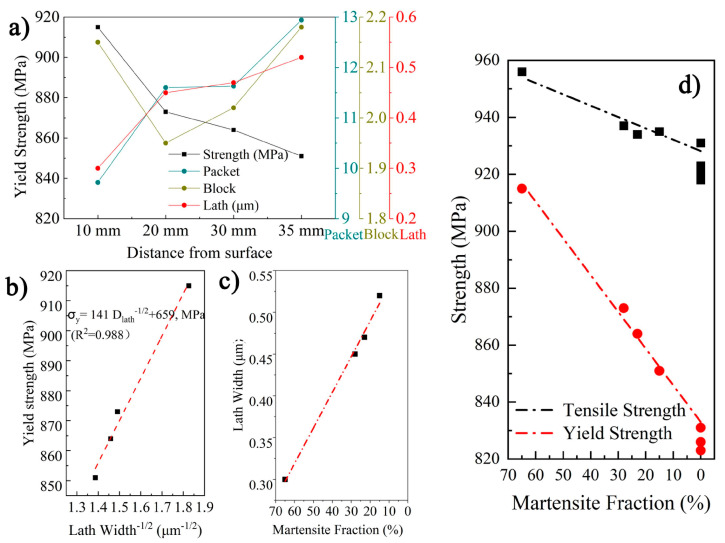
Correlation between microstructure and strength: (**a**) presents the correlation between M/LB substructure size and strength, (**b**) presents a linear relationship between lath width and strength, (**c**) presents a correlation between lath width and martensite fraction and (**d**) presents the correlation between martensite fraction and strength.

**Figure 11 materials-16-01607-f011:**
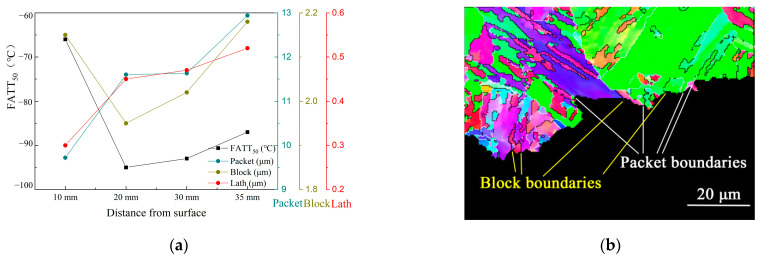
(**a**) Correlation between substructures size and toughness, and (**b**) IPF map selected from the impact sample profile shows an effect of the substructures’ boundary on the crack propagation.

**Figure 12 materials-16-01607-f012:**
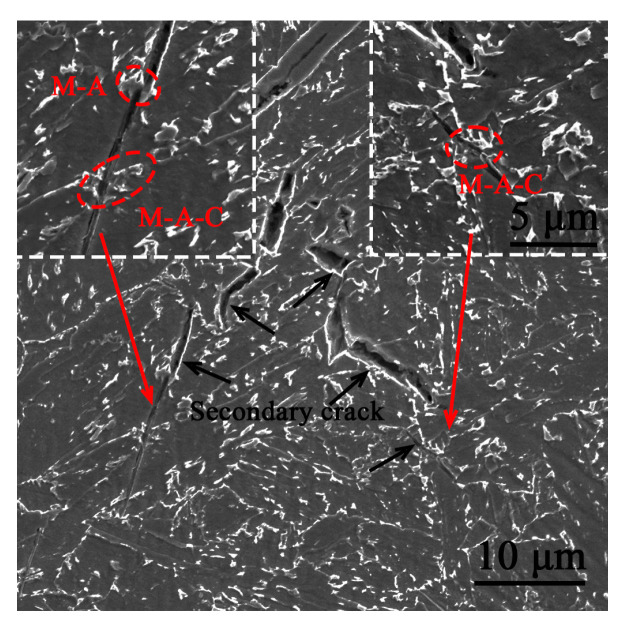
Secondary cracks were observed on the profile of the impact samples induced by broken M-A constituents.

**Figure 13 materials-16-01607-f013:**
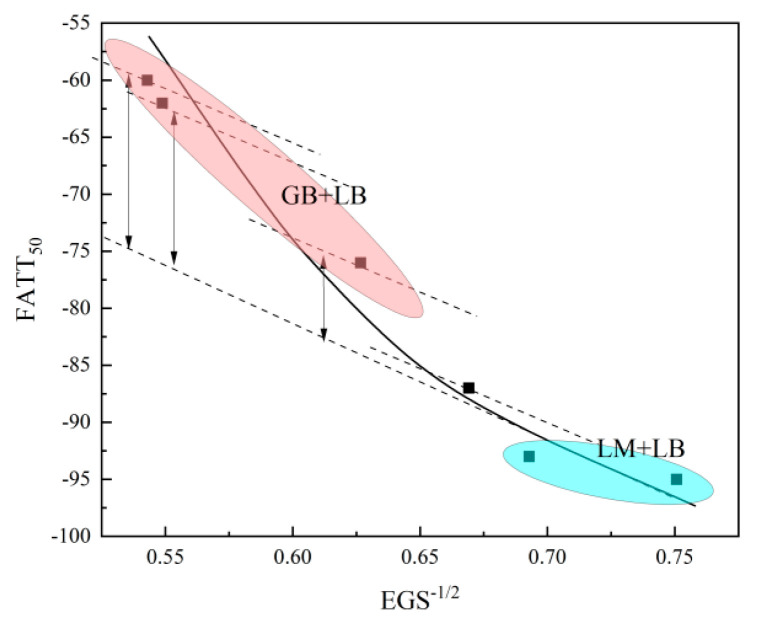
The correlation between the microstructure and FATT_50_. LM/LB fitted with the function of Hall–Petch (blue circle), and GB contributed to the rise of the FATT_50_ (red circle).

**Table 1 materials-16-01607-t001:** Chemical composition of the steel (wt%).

C	Si	Mn	Ni + Cr + Mo	V + Ti	Al	Fe
0.08	0.3	0.8	5	0.1	0.03	Base

**Table 2 materials-16-01607-t002:** Position of the specimens.

Specimens	L1	L2	L3	L4	L5	L6	L7
Distance from the surface (mm)	10	20	30	35	40	50	60

**Table 3 materials-16-01607-t003:** Quantitative results of the microstructure at different positions.

Location	AGS μm	Martensite Fraction	GB Fraction	M-A L_max_ μm	EGS (15°) μm	^1.^ Packet Size μm	^1.^ Block Width μm	Lath Width μm
10 mm	22	65%	/	/	2.26	9.72	2.15	0.30
20 mm	28	28%	/	/	1.77	11.60	1.95	0.45
30 mm	28	23%	/	/	2.08	11.63	2.02	0.47
35 mm	32	15%	6%	0.49	2.23	12.94	2.18	0.52
40 mm	38	/	14%	0.52	2.55	/	/	/
50 mm	42	/	20%	0.68	3.32	/	/	/
60 mm	45	/	32%	0.70	3.39	/	/	/

^1.^ These substructure units were not determined after 35 mm, where it has a large fraction of GB.

**Table 4 materials-16-01607-t004:** The deviation between the calculated value and experimental value.

Position	FATT_50_ (°C)	Calculated Value (°C)	EGS^−1/2^ (μm^−1/2^)	GB Fraction	MA Density (μm^−2^)	P(D_max_ > D_c_)	−Kd^1/2^ °C	FATT_50_ Increment by GB °C
10 mm	−66	−85	0.66	/	/	/	/	/
20 mm	−95	−96	0.75	/	/	/	−81	0
30 mm	−93	−90	0.69	/	/	/	−75	0
35 mm	−87	−86	0.67	6%	0.32	0.05	−72	1
40 mm	−76	−78	0.63	14%	0.38	0.09	−68	5
50 mm	−62	−67	0.66	20%	0.26	0.16	−59	7
60 mm	−59	−59	0.55	32%	0.27	0.18	−59	15

## Data Availability

No new data were created.

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
