# Peer review of "Effect of Microstructural Evolution on the Mechanical Properties of Ni-Cr-Mo Ultra-Heavy Steel Plate"

_materials, 2023, doi:10.3390/ma16041607_

Round 1

Reviewer 1 Report (New Reviewer)

The authors report about the relation between microstructure and mechanical properties of Ni-Cr-Mo steel plates. Unfortunately, the readability of the text is very limited by the poor English writing. The manuscript must be rewritten in correct and sound English writing.

This study yield a multivariate function between microstructure and the fracture appearance transition temperature. The effects of grain size, GB fraction, phase distribution, phase size on plate toughness are taken into account. It is not clear if the investigated samples stem from only one region of one band or from different regions or different bands.

Thus, it is questionable if the found relation is statistically relevant. A more detailed discussion of this statistical issue would be great.

Some minor issues are:

Table 1: physics' unit is missing
Fig. 2: scale is too small
Fig. 3: red letters show poor visibility on pics
Fig. 10: letter size too small

The manuscript is inacceptable regarding English spelling. A word processor correction should be a minimum to ensure an acceptable level of English writing.

Author Response

Dear reviewer:

We appreciate you taking the time to carefully read this article, and glad to receive your suggestions.

The language has been carefully corrected. The issues you point out has been revised.

The physics' unit in Table1 has been added.

The scale in Fig 2. has been enlarged.

The red letters in Fig 3.  has been  replaced by a white color

The fig 10. has been modified, and the letters has been enlarged

Besides, In this study, the investigated samples stem from different position across the thickness of an industrial heavy steel plate. However, the model is in accord with the current study. It's very meaningful for a comprehensive model to characterize the correlation between the microstructure and the low temperature toughness. In this study, what we intend to demonstrate is the evolution of microstructure and properties across the plate thicknesses, and it's difficult to collect more statistical data from the section of a 120 mm plate to modify the multivariate function.

Reviewer 2 Report (Previous Reviewer 2)

the authors have included my suggestions.

Author Response

Dear reviewer:

We appreciate you taking the time to carefully read this article, and glad to receive your suggestions.

Best wish for your regards

Reviewer 3 Report (Previous Reviewer 3)

The authors has corrected the paper and now warrants publication in Materials

Author Response

Dear reviewer:

We appreciate you taking the time to carefully read this article, and glad to receive your suggestions.

Best wish for your regards

Round 2

Reviewer 1 Report (New Reviewer)

English language can be improved. 

Author Response

Dear reviewer

We have proofread the whole manuscript and do a suitable improvement.

Kind regards,

This manuscript is a resubmission of an earlier submission. The following is a list of the peer review reports and author responses from that submission.

Round 1

Reviewer 1 Report

Authors must examine the the propriety of figure numbers in the text and the captions very carefully, specially Fig. 1 and Fig.2. 

Authors must learn the way to use the SI units, that there should be a space between the numerical value and the symbol.

Authors are strongly advised to acquire SAEDPs from the location where TEM images taken. Authors are required to achieve SAEDPs of where TEM images taken so that they could at least determine the microstructgures of carbides. In addition, the EDS should also be achieved to determine the compositions of them. I do not know the way authors determined the morphologies of carbides as rod-like without characterize them three-dimensionally.

In particular, the microstructures and compositions of carbides should be determined by SAEDPs and EDSs of carbides. Is the lath carbides the same as the rod-like carbides? How are the distribution of these carbides? Have authors measured the number density of these carbides? In addition, I understand the carbides seen in Fig. 3d was taken with dark-field condition, then authors are obliged to explain which particular electron diffraction spots of which carbides authors chose to acquire. Do they hold any orientation relationship with the matrix?

Authors should present the whole data of L1 to L7 for whole figures. The way they chose the images were ambiguous and not understandable.

What exactly is the physical meaning of L4, of this particular sample? Is it always 35 mm? Or, TEM images, SEM images, OM images of L4 are obviously necessary.

The way authors present the locations in figures are totally wrong, such as like location in Fig. 11(a), 12(a), they are obliged to use 10, 20, 30, 35 mm, instead of L1, L2, L3 and L4, with physical distances.

At last, Authors are strongly advised to present the microstructural data of the whole samples, not the ones authors find easy to explain.

Reviewer 2 Report

The authors have carried out considerable characterization and quantitative analysis to arrive at their conclusions. The conclusions are typically what one would expect.  However, they made several avoidable mistakes, which make it difficult to read the paper.

Here are a few examples.:

They mention Fig 18 in the text (line 311). There is no figure 18 in the manuscript. Similarly Fig 10a (line 307)

They mention the author but do not give the reference. 

Figure 11b caption do not describe the figure. 

Caption of Fig 8c does not describe the figure

They should check the paper thoroughly for similar mistakes.

Reviewer 3 Report

The paper presents a lot of experimental work. Unfortunately most part of this work is presented in a very poor way. The English sentences in many cases are simply unintelliglible. Several sentences have no sublective and objective, sometimes it is simply impossible to understand what the authors wanted to tell. Many abbreviations are not defined or not clearly defined. In the Discussion section the reader can hardly separate the original results and the results from the literature. Overall summary: the paper in the present form should not be accepted. The authors should re-write the whole paper, asking some native English speaker to check the English, and present clearly their own and new results.